# Phenotyping Rare CFTR Mutations Reveal Functional Expression Defects Restored by TRIKAFTA^TM^

**DOI:** 10.3390/jpm11040301

**Published:** 2021-04-15

**Authors:** Onofrio Laselva, Maria C. Ardelean, Christine E. Bear

**Affiliations:** 1Programme in Molecular Medicine, Hospital for Sick Children, Toronto, ON M5G 8X4, Canada; onofriolaselva@gmail.com (O.L.); maria.ardelean.17@ucl.ac.uk (M.C.A.); 2Department of Medical and Surgical Sciences, University of Foggia, 71122 Foggia, Italy; 3Department of Neuroscience, Physiology and Pharmacology, University College London, London WC1E 6BT, UK; 4Department of Physiology, University of Toronto, Toronto, ON M5G 8X4, Canada; 5Department of Biochemistry, University of Toronto, Toronto, ON M5G 8X4, Canada

**Keywords:** cystic fibrosis, CFTR, TRIKAFTA, rare mutation, H609R, I1023_V1024del

## Abstract

The rare Cystic Fibrosis Transmembrane Conductance Regulator (CFTR) mutations, c.1826A > G (H609R) and c.3067_3072delATAGTG (I1023_V1024del), are associated with severe lung disease. Despite the existence of four CFTR targeted therapies, none have been approved for individuals with these mutations because the associated molecular defects were not known. In this study we examined the consequences of these mutations on protein processing and channel function in HEK293 cells. We found that, similar to F508del, H609R and I1023_V1024del-CFTR exhibited reduced protein processing and altered channel function. Because the I1023_V1024del mutation can be linked with the mutation, I148T, we also examined the protein conferred by transfection of a plasmid bearing both mutations. Interestingly, together with I148T, there was no further reduction in channel function exhibited by I1023-V1024del. Both H609R and I1023_V1024del failed to exhibit significant correction of their functional expression with lumacaftor and ivacaftor. In contrast, the triple modulator combination found in TRIKAFTA^TM^, i.e., tezacaftor, elexacaftor and ivacaftor rescued trafficking and function of both of these mutants. These in-vitro findings suggest that patients harbouring H609R or I1023_V1024del, alone or with I148T, may benefit clinically from treatment with TRIKAFTA^TM^.

## 1. Introduction

Cystic Fibrosis (CF) is caused by mutations in the cystic fibrosis transmembrane conductance regulator (*CFTR*) gene. The CFTR protein is an ion channel that mediates chloride and bicarbonate transport in the epithelial cells of multiple organs including the lungs, pancreas and intestine [1,2,3]. More than 2000 mutations have been identified in CFTR (CFTR1, www.genet.sickkids.on.ca (accessed on 14 April 2021)). The primary defects caused by some of these mutations, have been identified [4]. The major mutation F508del causes misfolding and mistrafficking of the CFTR protein, and these defects can be partially rescued by small molecule modulator combinations that correct the trafficking defect and enhance CFTR channel activity at the cell surface. The combination lumacaftor (VX-809) with ivacaftor (VX-770) has been approved by the US Food and Drug Administration (FDA) as Orkambi^®^ for patients bearing the F508del mutation, and more recently, the related tezacaftor (VX-661)-ivacaftor (VX-770) combination has been approved as Symdeko^®^ [5,6]. The triple combination of two corrector molecules, elexacaftor (VX-445) and tezacaftor (VX-661), together with the potentiator ivacaftor (VX-770), was approved as TRIKAFTA^TM^ for the treatment of patients bearing F508del on at least one allele [7]. 

Recently, the FDA agreed to extend the list of CF-causing mutations for which TRIKAFTA^TM^ treatment could be clinically beneficial based on in-vitro, cell-based studies [8]. To be included on the extended list, recombinant mutant proteins, expressed in a heterologous expression system, needed to show an increase in CFTR chloride channel function in response to TRIKAFTA^TM^ treatment [8]. Although there are few patients harbouring the rare mutations, c.1826A > G (H609R) or c.3067_3072delATAGTG (I1023_V1024del), their need for effective therapy is urgent. Therefore, we were encouraged by the FDA decision to study their in-vitro response to TRIKAFTA^TM^. 

The H609R mutation is found in the Ecuadorian population and was found in 9 patients in the database [4]. I1023_V1024del-CFTR (legacy name of 3199del6), is a rare CF-causing mutation that is relatively frequent in the French-Canadian population [9] and caused by in-frame deletion of six base pairs in exon 17a. This mutation was previously identified in association with I148T in CF patients [9,10]. CF patients bearing H609R and I1023_V1024del-CFTR mutations show pancreatic insufficiency, reduction of ppFEV1 and abnormal Sweat Cl (>60 Eq/L) [11,12,13]. While it has been demonstrated that I148T showed WT-CFTR level functional expression [4,14]), there are no functional studies on I1023_V1024del-CFTR alone or in combination with I148T.

In the present study, we first investigated the molecular consequences of these two rare CF-causing mutations, H609R and [I148T/1023_V1024del]-CFTR, and the efficacy of TRIKAFTA^TM^ in restoring function of these variants in HEK-293 cells. The trafficking and conductance defects exhibited by these mutants in our biochemical and functional studies suggest that such mutations might be classified as class II mutations. We also show that H609R and I1023_V1024del can be rescued by TRIKAFTA^TM^. 

## 2. Material and Methods

### 2.1. Cell Culture and Transfection 

Human embryonic kidney GripTite™ cells (HEK293) were maintained as previously described [15]. CFTR mutants were generated by AGCT corporation (Toronto, ON, Canada) containing WT-CFTR cDNA (in pcDNA3.1) as the template as previously described [16]. HEK293 cells were transiently transfected using PolyFect Transfection Reagent (Qiagen, Hilden, Germany), as previously described [17].

### 2.2. CFTR Channel Function in HEK Cells

HEK293 cells were transiently transfected with WT, F508del, H609R, I1023_V1024del-CFTR or the [I148T; I1023_V1024del]-CFTR constructs. After18h, the cells were treated with DMSO (0.1%), 3 µM VX-809, and 3 µM VX-661 + 3 µM VX-445 (Selleck Chemicals, Houston, TX, USA) for 24 h at 37 °C. Then, the cells were loaded with a blue membrane potential dye dissolved in chloride free buffer [18] for 35 min at 37 °C. The plate was then read in a fluorescence plate reader (SpectraMax i3; Molecular Devices, San Jose, CA, USA) at 37 °C. After 5 min baseline, CFTR was stimulated using forskolin (10 µM; Sigma-Aldrich, St. Louis, MO, USA) and the potentiator VX-770 (1 µM, Selleck Chemicals, Houston, TX, USA). Then, CFTR inhibitor (CFTRinh-172, 10 µM, Cystic Fibrosis Foundation Therapeutics, Bethesda, MD, USA) was added to deactivate CFTR. The peak changes in fluorescence to CFTR agonists were normalized relative to the fluorescence recorded immediately before the addition of forskolin [19,20].

### 2.3. Immunoblotting

HEK293 were lysed in modified radioimmunoprecipitation assay (RIPA) buffer (50 mM Tris-HCl, 150 mM NaCl, 1 mM EDTA, pH 7.4, 0.2% SDS, and 0.1% Triton X-100) with protease inhibitor cocktail (Roche, Mannheim, Germany) for 10 min in ice [21]. Soluble fractions were analyzed by SDS-PAGE on 6% Tris-Glycine gels (Life Technologies, Carlsbad, CA, USA). After electrophoresis, proteins were transferred to nitrocellulose membranes (Bio-Rad, Hercules, CA, USA) and incubated in 5% milk. CFTR bands were detected with human CFTR-specific murine mAb 596 (1:5000 dilution, UNC, Chapel Hill, NC, USA) and Calnexin with human Calnexin-specific rabbit (1:10,000, Sigma-Aldrich, St. Louis, MO, USA). The blots were developed with ECL (Bio-Rad, Hercules, California, USA) using the Li-Cor Odyssey Fc (LI-COR Biosciences, Lincoln, NE, USA) in a linear range of exposure (0.5–2 min) [22,23]. CFTR and Calnexin proteins level were quantified by densitometry of immunoblots using ImageJ (NIH, Bethesda, MD, USA) [24,25]. The non-specific signal was removed by subtracting the background from both bands C and B. 

## 3. Statistical Analysis

Data are represented as mean ± S.D. GraphPad Prism 8.0 software (San Diego, CA, USA) was used for all statistical tests. One-way ANOVA was conducted and *p*-values < 0.05 were considered significant. Data with multiple comparison were assessed with Tukey’s multiple-comparison test with α = 0.05. 

## 4. Results

In order to study the molecular consequences of the H609R and I1023_V1024del mutations, we used HEK293 cells to overexpress those mutations for biochemical and functional studies. We also determined whether existing therapies were effective in improving their function, thus providing insights into personalized medicine for CF patients. CFTR channel function was measured using the membrane depolarization assay (FLIPR) and we measured the CFTR chloride channel response the addition of forskolin and correctors/potentiators, as well as inhibitors. 

Recently, we showed that VX-661 + VX-445 treatment significantly rescued F508del-CFTR channel activity and protein processing to close to fifty-percent of Wt-CFTR, in HEK293 and primary nasal epithelial cells [26]. Therefore, we used the Wt-CFTR and rescued (VX-661 + VX-445 + VX-770) F508del-CFTR processing and channel activities as benchmarks with which to evaluate the effect of the CFTR modulators on H609R and I1023_V1024del-CFTR. 

Western blot studies demonstrated that H609R, I1023_V1024del and I148T/I1023_V1024del mutations induced a folding defect by diminishing the relative amount of complex CFTR produced (Band C/Band B+ Band C i.e., C/C + B) in comparison to this ratio measured for Wt-CFTR (Figure 1A,B). Relative to un-treated samples, we detected a significant increase in the ratio of C/C + B after 24 h treatment with VX-809 or VX-661 + VX-445 for all CFTR mutants: F508del, H609R, I1023-V1024Vdel and the compound mutant: I148T plus I1023_V1024del. As expected, the combination of VX-661 + VX-445 showed superior rescue of processing for all of these mutations compare the single treatment with VX-809. Interestingly, we found that the compound mutant: I148T plus I1023_V1024del exhibited less of a processing rescue response to VX-661 + VX-445 than I1023_V1024del protein alone (Figure 1A,B). 

As shown in Figure 2A,B,D we found that forskolin-dependent H609R and I1023_V1024del-CFTR channel function was not improved after a 24 h treatment with VX-809 and acute potentiation with VX-770. Interestingly, chronic treatment with VX-661 + VX-445 did rescue VX-770-potentiated activity of H609R and I1023_V1024del-CFTR in HEK293 cells and this is approximately 95% of the VX-770 potentiated F508del-CFTR channel activity rescued by VX-661 + VX-445. Since we did not see functional rescue after correction with VX-809 alone and VX-661 behaves similarly to VX-809 [26], we suggest that the VX-445 is required to rescue the functional expression of these mutants.

Since the I148T mutation can be linked with I1023_V1024del [3199del6] in CF patients [27], we then investigated the efficacy of VX-661 + VX-445 in rescuing the compound mutant: I148T/I1023_V1024del-CFTR. Importantly, we did see a significant increase of I148T/I1023_V1024del-CFTR function after acute potentiation with VX-770 and chronic treatment with VX-661 + VX-445 (Figure 2D).

## 5. Discussion

In this study we showed that the defective protein processing and channel activity caused by the H609R and I1023_V1024del mutations could be partially rescued by the FDA-approved drug combination of tezacaftor (VX-661), elexacaftor (VX-445) and ivacaftor (VX-770; TRIKAFTA^TM^).

I1023_V1024del mutation can occur in association with the MSD1-localized missense mutation, I148T [9,10]. Interestingly, CF patients bearing I148T are typically asymptomatic with normal sweat chloride and ppFEV1 [11]. Choi et al., further demonstrated that I148T-CFTR showed normal protein processing and channel activity like WT-CFTR in HEK293 cells [14]. Moreover, Terlizzi et al., measured that the CFTR channel activity of I148T-CFTR is 87% of WT in primary nasal epithelial (HNE) cells from two heterozygous donors with the I148T mutation [11]. On the other hand, [I148T; I1023_V1024del]-CFTR expressed in the HNE cells of 3 CF patients which were heterozygous with minimal function class II mutations showed only approximately 7% of WT-CFTR activity [11]. Such studies suggest that it is the presence of I1023_V1024del-CFTR, not I148T leads to CF [27,28]. 

As shown in Figure 1 and Figure 2, we showed that the I1023_V1024del-CFTR protein is misprocessed and exhibits reduced function. Further, the functional defect, at least in HEK-293 cells is not further reduced in the compound mutant: I148T/I1023_V1024del. The triple combination of modulators in TRIKAFTA, partially rescues processing and functional defects both mutations. However, we recognize the need to corroborate these findings and probe the mechanism of drug action in relevant airway epithelial cells. It has been shown that protein processing is dependent on the host cell type [29] and we hypothesize that the peripheral quality control mechanisms regulating membrane protein stability will also depend on the cellular context. Unfortunately, the proportion of CF patients bearing both I1023_V1024del and I148T is only 0.9% of the patient population (0.07% of CF mutations) [30]; hence, access to tissues from patients who are homozygous for compound mutations is somewhat limited. 

H609R is an R domain localized missense mutation highly prevalent in the CF population of Ecuador (19%) [12,31,32]. Moya et al. discovered four cases of homozygous CF patients expressing the mutation H609R-CFTR. Clinical data from these four patients showed a typical CF phenotype (Sweat Cl > 60 mEq/L, Pancreatic insufficiency, ppFEV 36–75% and colonization by *Pseudomonas aeruginosa)* [12]. Furthermore, functional studies of H609R-CFTR in Fischer rat thyroid cell line models showed a residual function and a minimal response to VX-809 and VX-770 (ORKAMBI) [33]. As expected, the H609R mutation caused defects in protein processing in the HEK-293 expression system (Figure 1). The molecular basis for the deleterious effect of H609R on processing remains unknown, but it is reminiscent of our previous studies that showed misprocessing of L610S-CFTR [34]. Both missense mutations are localized in the Regulatory Domain (RD), a region that may interact 14-3-3 protein [35] and has been implicated in CFTR trafficking through the biosynthetic compartment. Misprocessing of H609R-CFTR may be conferred by altered interactions with the 14-3-3 protein and we will test this idea in our future work. 

Interestingly, unlike F508del-CFTR, there was no significant potentiated activity conferred by VX-770 and forskolin in the case of the H609R and I1023_V1024del mutant proteins, despite partial rescue by VX-809 of their processing defects. These findings suggest that there are functional defects conferred by H609R and I1023_V1024del. These functional defects were partially repaired following treatment with the triple combination, suggesting that the novel modulator: VX-445 exerts a distinct effect that enables superior functional rescue for the H609R and I1023_V1024del mutants. 

Overall, the magnitude of the functional rescue with VX-661, VX-445 and VX-770 for both of these mutations, H609R and I1023_V1024del is approximately 50% of WT-CFTR and is comparable to the activity of rescued F508del-CFTR recorded via membrane depolarization assays in HEK293 cells (Figure 2). Such findings suggest that TRIKAFTA treatment may cause a significant clinical response in patients bearing these mutations. On the other hand, the clinical response size is highly variable even in patients homozygous for F508del [7]. Therefore, we and others are examining the utility of modulator testing on patient-derived nasal cultures or rectal organoids to predict clinical response size [18,36,37,38,39,40,41,42,43,44,45,46,47]. 

## Figures and Tables

**Figure 1 jpm-11-00301-f001:**
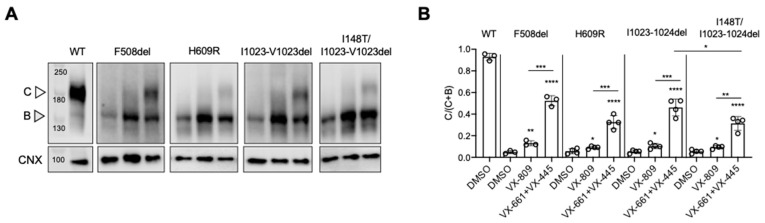
H609R and I1023_V1024del-CFTR exhibit protein processing defects and is restored by VX-661 + VX-445. (**A**) Immunoblots of steady-state expression of WT, F508del, H609R, I1023_V1024del, I148T/I1023_V1024-CFTR and treated for 24 h at 37 °C with: DMSO (0.1%), 3 μM VX-809 and 3 μM VX-661 + 3 μM VX-445. C: mature, complex-glycosylated CFTR; B: immature, core-glycosylated CFTR; CNX, Calnexin as loading control. (**B**) Bars represent the mean (±SD) of the ratio C/(C + B) (*n* = 3–5) (* *p* < 0.05; ** *p* < 0.01; *** *p* < 0.001; **** *p* < 0.0001).

**Figure 2 jpm-11-00301-f002:**
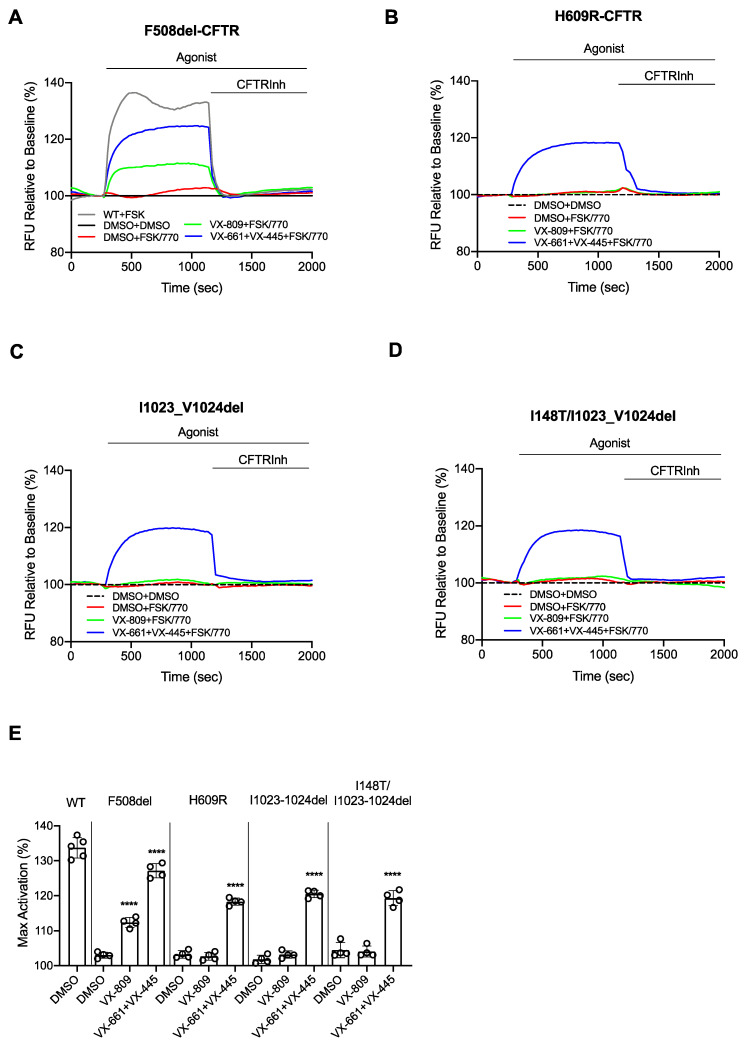
H609R and I1023_V1024del-CFTR are rescued by VX-661 + VX-445 + VX-770. HEK293 cells were transiently transfected with WT, F508del, H609R, I1023_V1024del, I148T/I1023_V1024-CFTR and treated for 24 h at 37 °C with: DMSO (0.1%), 3 μM VX-809 and 3 μM VX-661 + 3 μM VX-445. Representative traces of (**A**) WT and F508del, (**B**) H609R, (**C**) I1023_V1024del, (**D**) [I148T; I1023_V1024del]-CFTR function (membrane depolarization assay) in HEK293 cells. (**E**) Bar graphs show the mean (±SD) of maximal activation of mutated CFTR after stimulation by 10 µM FSK +1 µM VX-770 in H609R, I1023_V1024del, I148T/I1023_V1024-CFTR HEK293 cells and 10 µM FSK in WT-CFTR HEK293 cells (*n* = 4 biological replicates and 4 technical replicates for each experiment). (**** *p* < 0.0001).

## Data Availability

The study did not report any data.

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
