# Peer review of "Phenotyping Rare CFTR Mutations Reveal Functional Expression Defects Restored by TRIKAFTATM"

_jpm, 2021, doi:10.3390/jpm11040301_

Round 1

Reviewer 1 Report

The authors report that CFTR mutations H609R and del-I1023_V1024 cause processing and gating defects upon heterologous expression in Hek293 cells that can be ameliorated with the Trikafta corrector/potentiator-combination. The manuscript is generally well developed and should be of interest to the CF community. Specific points to improve the manuscript are detailed below:

  • Fig.1. is missing critical reference points; principally WT-CFTR and dF508-CFTR immunoblots. Rather than referencing data published elsewhere, the authors should show immunoblots (and IB quantification) for both WT and dF508-CFTR treated with the same drugs (DMSO, VX809, VX661+VX445).
  • Fig. 1: More importantly, rather than just reporting the ratios of band C over B+C, the authors should also quantify/compare the total amount of CFTR protein expressed for these five CFTR constructs to answer two questions: 1. What is the effect of drug treatment on total CFTR protein levels for each CFTR variant (as could be quantified for three CFTR variants from the blots shown in the current Fig. 1A); and 2. Is total CFTR protein expression different between WT, dF508, and the three novel constructs (to this end, the different CFTR variants would have to be loaded onto the same gel/membrane). The latter will have critical impact upon the interpretation of the activity data shown in Fig. 2.
  • Fig. 2. Please include data for dF508-CFTR treated with DMSO, VX-809 or VX661-VX45 in Fig. 2D.
  • Fig. 2. For H609R and del-I1023_V1024, do the authors know whether VX661 or VX445 exerts the predominant effect on rescuing CFTR function, or whether their combination is critical to produce an effect?
  • dF508 produces both processing and gating defects. The data in Fig. 2 clearly suggest processing defects also for H609R and del-I1023_V1024. However, it is not clear whether these mutations also induce a gating defect. To this end, the authors should compare CFTR activity after VX661+VX445 treatment in the absence of presence of the potentiator VX770.
  • In the Methods section, the authors state that data represent Mean +/- SEM, in the figure legends the authors state that data represent Mean +/- SD.
  • Please rephrase Ln 91/92 “…membrane potential dye FOR dissolved…”
  • Ln 96: please delete “THE” from “…using THE forskolin…”

Author Response

The authors report that CFTR mutations H609R and del-I1023_V1024 cause processing and gating defects upon heterologous expression in Hek293 cells that can be ameliorated with the Trikafta corrector/potentiator-combination. The manuscript is generally well developed and should be of interest to the CF community. Specific points to improve the manuscript are detailed below:Fig.1. is missing critical reference points; principally WT-CFTR and dF508-CFTR immunoblots. Rather than referencing data published elsewhere, the authors should show immunoblots (and IB quantification) for both WT and dF508-CFTR treated with the same drugs (DMSO, VX809, VX661+VX445).

We agree with our reviewer and we includethese new experimental data in our revision.

Fig. 1: More importantly, rather than just reporting the ratios of bandC over B+C, the authors should also quantify/compare the total amount of CFTR protein expressed for these five CFTR constructs to answer two questions: 1. What is the effect of drug treatment on total CFTR protein levels for each CFTR variant (as could bequantified for three CFTR variants from the blots shown in the current Fig. 1A); and 2. Is total CFTR protein expression different between WT, dF508, and the three novel constructs(to thisend, the different CFTR variants would have to be loaded onto thesame gel/membrane). The latter will have critical impact upon the interpretation of the activity data shown in Fig. 2.

The goal of our work was to determine the consequencesof two rare disease-causing mutations on the processing and channel activity of CFTR and to determine if the triple modulator combination(TRIKAFTA) was effective in rescuing these defects. Towards this goal, we studiedmodulator responses relative to theresidual expression and function of each mutant. In this way, each mutantserves as its own control when assessing modulator activity.We discussed these analyses in the appropriate results section (lines 129-134). We conducted our analysis this way because variabilitiesin total protein expressioncould be due to experimental factors such differential transfection efficiencies.

Fig. 2. Please include data for dF508-CFTR treated with DMSO, VX-809 or VX661-VX45in Fig. 2D.We included thenew, requested datain .

Fig. 2. For H609R and del-I1023_V1024, dothe authors know whether VX661 or VX445 exerts the predominant effect on rescuing CFTR function, or whether their combination is critical to produce an effect?

Since we dont see functional rescue after correction with VX-809aloneandVX-661 behavessimilarlyto VX-809(Laselva et al., European Respiratory Journal, 2020), we suggest thatthe VX-445 is required to rescue the functional expression of these mutants. We revisedthe manuscript(lines 149-151)to highlightthis point.

The data in Fig. 2 clearly suggest processing defects also for H609R and del-I1023_V1024. However, it is not clear whether these mutations also induce a gating defect. To this end, the authors should compare CFTR activity after VX661+VX445 treatment in the absence of presence of the potentiatorVX770.

We revised the manuscript in order to highlight our findingthat both H609R and I1023_V1024del exhibit defective channel activity(lines 149-151,194-198).Functionaldefects wereapparent in our studies of the effect of lumacaftor plus ivacaftor on these mutants. Unlike F508del-CFTR, there was no significant potentiated activity conferredby ivacaftor and forskolinin the case of theH609R and I1023_V1024delmutantproteins, despite rescueby lumacaftor of their processing defect. These findings show that there arefunctional defects conferred by H609R and I1023_V1024del. Interestingly, the functional defect(s) were repaired following treatment with the triple combination, suggesting that VX-445 exerts a distinct effectthat enables superior functional rescue for theH609R and I1023_V1024delmutants.

In the Methods section, the authors state that data represent Mean +/-SEM, in the figure legends the authors state that data represent Mean +/-SD.

We revised the manuscriptto correct this. Thank you.

Please rephrase Ln 91/92 “...membrane potential dye FOR dissolved...”

We revised the text as suggested.

Ln 96: please delete “THE” from “...using THE forskolin...”

We changed as suggested

Reviewer 2 Report

In this work, the authors have shown FDA-approved drug combination of tezacaftor 168 (VX-661), elexacaftor (VX-445) and ivacaftor (VX-770; TRIKAFTATM) was able to correct some CF-associated rare mutations such as H609R, I1023_V1024del, and I148T/I1023_V1024del. The work is well presented and the experiments are shown in a logical way. However, some key aspects have to be addressed before being considered for publication.

Major point:

  • In figure 1, the authors have shown the effect of drug treatment on CFTR mutants transiently transfected in HEK293 cells. The rescued effect was analyzed by looking at maturation on the proteins and in particular the appearance of the Band C which corresponds to the complex-glycosylated form achieved once the proteins have trafficked through the Golgi complex. The overexpression condition does not allow the correct estimation of the protein maturation. Indeed, under the overexpression condition, most of the exogenous proteins could not receive the proper modification of the glycans by the Golgi enzymes leading to the underestimation of the abundance of Band C with respect to the accumulation of Band B. Thus, this experiment has to be performed in stably-expressing cells or, even better, in primary cells from patients. Furthermore, as clearly shown by the blots, upon treatment with VX-809 the protein level of all three mutants appeared increased as a logical consequence of the treatment with a chemical chaperon which exerts its effect stabilizing its target. This effect was evident also with the combination of VX-661+VX-445 which does not only stabilize the protein but favors also its maturation. The authors should discuss it providing a possible explanation. Moreover, the lack of a control (CFTR-WT treated or not with the indicated drugs) does not allow to estimate of the real entity of the rescue (% of the mature form with the respect to the WT used as reference). Last, the quantification shown in the graph on the right does not reflect the result shown in the western blotting on the left. Indeed, the graph reports the ratio between the amount of Band C with respect to the total (B+C). For all three CFTR mutants, upon treatment with VX-661+VX-445, the value of that ratio is shown to be between 0.6 and 0.8 (60-80%). However, these values are not mirrored by the blots. The authors have to clarify this incongruence.
  • In figure 2 lacks the graph relative to the CFTR-WT which is reported in the quantification (D). Moreover, the authors have to report also the effect of the drug treatment on the CFTR-WT and not only the mutants. This will provide also important information about the drug effects as either stabilizing or correcting agents.

Author Response

In this work, the authors have shown FDA-approved drug combination of tezacaftor 168 (VX-661), elexacaftor (VX-445) and ivacaftor (VX-770; TRIKAFTATM) was able to correct some CF-associated rare mutations such as H609R, I1023_V1024del, and I148T/I1023_V1024del. The work is well presented and the experiments are shown inalogical way. However, some key aspects have to be addressed before being considered for publication.Major point:In figure 1, the authors have shown the effect ofdrugtreatment on CFTR mutants transiently transfected in HEK293 cells. The rescued effect wasanalyzed by looking at maturation on the proteins and in particular the appearance of the Band C which corresponds to the complex-glycosylated form achieved once theproteins have trafficked through the Golgi complex. The overexpression condition does notallow the correct estimation of the protein maturation. Indeed, under the overexpression condition, most of the exogenous proteins could not receive the proper modification of the glycans by the Golgi enzymes leading to the underestimation of the abundance of Band C with respect to the accumulation of Band B. Thus, this experiment has to be performed in stably-expressing cells or, even better, in primary cells from patients.

We revised our discussion to includethe important caveats to studying mutantproteins in transient overexpression systemsraised by our reviewer(lines 190-198). Indeed, such a caveat wouldapply to the studies of mutant CFTRprotein stably expressedinHEK-293 cells. As ourreviewer mentioned, it has been shown that there is an impact of the host cell type on glycan modification(Goh et al, Cristical Reviews in Biotechnology, 2017) and we includereference to this particular caveat on pages 192-195of our revised manuscript. Futurestudiesincluding biochemical studies of primary tissue cultures from theseindividualsare much more relevant and we will be planned when we can obtain such cultures.

Furthermore, as clearly shown bythe blots, upontreatment with VX-809 the protein level of all threemutantsappeared increased as a logical consequence of the treatment with a chemicalchaperon which exerts its effect stabilizing its target. This effect was evident also with the combination of VX-661+VX-445 which does not only stabilize the protein but favors also its maturation. The authors should discuss it providing a possible explanation.

We revised our manuscript to discuss thepossibility that the combination: VX-661 and VX-445 may act to repairprocessing and to enhance stability of these rare mutants atthe cell surface. Please see lines190-198.Moreover, the lack of a control (CFTR-WT treated or not with the indicated drugs) does not allow to estimate of the real entity of the rescue (% of the mature form with the respect to the WT used as reference).

We include new studies of Wt-CFTR and F508del-CFTR processing and function inour revision.Processing for the WT-CFTR(C/C+B) is close to one in our new experiment (recently added to figure 1) and as reported in multiplepublications. Themutant studied here exhibit residual C/C+B values of close to 0.2,Hence,they are clearly misprocessed. We observed statistically significant increases in the C/C+B ratios for each mutant for VX-809 and VX-445+VX-661.

Last, the quantification shown in the graph on the right does not reflect the result shown in the westernblotting on the left. Indeed, the graph reports the ratio between the amountof Band C with respect to thetotal (B+C). For all threeCFTRmutants, upon treatment with VX-661+VX-445, thevalue ofthat ratio is shown to be between 0.6 and 0.8 (60-80%). However, these values are not mirrored by the blots. The authors have to clarifythis incongruence.

In response to our reviewers comments,were-analyzedthe data using ImageJ softwareand these newanalyses are included in the revised Figure 1.

In figure 2 lacks the graph relative to theCFTR-WTwhich is reported in the quantification (D). Moreover, the authors have to report also the effectof the drug treatment on the CFTR-WT and not only the mutants. This will provide also important information about the drug effects as either stabilizing or correcting agents.

Wt-CFTR channel function is now shown(in our revised manuscript) in panel2E. As our reviewer suggests, detailed studies of themechanism of action of these modulators should include examination of their effect on wild type CFTR proteinand mutant CFTR proteins.However, the purpose of the current study wasto evaluate the potential for in-vitro efficacy of CFTR modulators fortwo,understudied mutants. Future, in-depthstudies of the mechanism of action of VX-661+VX-445 are being planned and will include: Wt-CFTR.

Round 2

Reviewer 1 Report

the manuscript is acceptable for publication with the revised figure.

Author Response

We thanks the reviewer #1

Reviewer 2 Report

They correct the blots in the new figure 1 (this is appreciable) but the problem relative to the quantification still remains. If we look again at the bands' intensities in the blots and we try to compare them with the values reported in the graph we can found correspondence only for the WT. Just to make a more concrete example, the authors have measured the abundance of the mature form of CFTR (Band C for either WT or mutants before and after treatment with drugs) relative to the total amount of the protein for each line (Band B+C). If we look at the value relative to the WT we can find a nice correspondence since the amount of the Band C is actually at least 90% of the total (B+C). However, if we look at the values relative to each mutant before treatment with drugs, we found that for each mutant the Band C is about 20% of the total but if we look at the blots we do not find any trace of those forms. Thus, I suspect that the authors have made the calculation without subtracting the background signal for each measurement and, as such, the final values appear overestimated and do not show adequate correspondence with the blots shown in the same figure. This has to be corrected for each panel before considering it for publication.

Author Response

Reviewer #2

They correct the blots in the new figure 1 (this is appreciable) but the problem relative to the quantification still remains. If we look again at the bands' intensities in the blots and we try to compare them with the values reported in the graph we can found correspondence only for the WT. Just to make a more concrete example, the authors have measured the abundance of the mature form of CFTR (Band C for either WT or mutants before and after treatment with drugs) relative to the total amount of the protein for each line (Band B+C). If we look at the value relative to the WT we can find a nice correspondence since the amount of the Band C is actually at least 90% of the total (B+C). However, if we look at the values relative to each mutant before treatment with drugs, we found that for each mutant the Band C is about 20% of the total but if we look at the blots we do not find any trace of those forms. Thus, I suspect that the authors have made the calculation without subtracting the background signal for each measurement and, as such, the final values appear overestimated and do not show adequate correspondence with the blots shown in the same figure. This has to be corrected for each panel before considering it for publication.

We thank our reviewer. As suggested, the immunoblots have been reanalyzed with background signal subtraction and the new data are presented in the revised Figure 1B.  The Methods and Material section has also been revised to better discuss the western blot analysis methods on lines 102-103.
